# sumSTAAR: A flexible framework for gene-based association studies using GWAS summary statistics

**Nadezhda M. Belonogova**[1]*, **Gulnara R. Svishcheva**[1,2], **Anatoly V. Kirichenko**[1], **Irina V. Zorkoltseva**[1], **Yakov A. Tsepilov**[1,3], **Tatiana I. Axenovich**[1]*

**1** Laboratory of Segregation and Recombination Analyses, Institute of Cytology and Genetics, Siberian Branch of the Russian Academy of Sciences, Novosibirsk, Russia, **2** Laboratory of Animal Genetics, Vavilov Institute of General Genetics, the Russian Academy of Sciences, Moscow, Russia, **3** Department of Natural Sciences, Novosibirsk State University, Novosibirsk, Russia

☯ These authors contributed equally to this work.
* belon@bionet.nsc.ru (NMB); tatiana.aksenovich@gmail.com (TIA)

**Data Availability Statement:** sumSTAAR is the part of sumFREGAT package (function sumSTAAR()) sumFREGAT v.1.2.3: https://CRAN.R-project.

## Abstract

Gene-based association analysis is an effective gene-mapping tool. Many gene-based methods have been proposed recently. However, their power depends on the underlying genetic architecture, which is rarely known in complex traits, and so it is likely that a combination of such methods could serve as a universal approach. Several frameworks combining different gene-based methods have been developed. However, they all imply a fixed set of methods, weights and functional annotations. Moreover, most of them use individual phenotypes and genotypes as input data. Here, we introduce sumSTAAR, a framework for gene-based association analysis using summary statistics obtained from genome-wide association studies (GWAS). It is an extended and modified version of STAAR framework proposed by Li and colleagues in 2020. The sumSTAAR framework offers a wider range of gene-based methods to combine. It allows the user to arbitrarily define a set of these methods, weighting functions and probabilities of genetic variants being causal. The methods used in the framework were adapted to analyse genes with large number of SNPs to decrease the running time. The framework includes the polygene pruning procedure to guard against the influence of the strong GWAS signals outside the gene. We also present new improved matrices of correlations between the genotypes of variants within genes. These matrices estimated on a sample of 265,000 individuals are a state-of-the-art replacement of widely used matrices based on the 1000 Genomes Project data.

## Author summary

Gene-based association analysis is an effective gene mapping tool. Quite a few frameworks have been proposed recently for gene-based association analysis using a combination of different methods. However, all of these frameworks have at least one of the disadvantages: they use a fixed set of methods, they cannot use functional annotations, or they use

org/package=sumFREGAT LD matrices for imputed genotypes from UKBB sample: https://mga.bionet. nsc.ru/sumfregat/ukbb/ LD matrices for exome sequences from UKBB sample: https://mga.bionet. nsc.ru/ukbb_exome_matrix/ Polygene pruning: https://github.com/nbelon/Polygene_pruning Functional annotations from FAVOR v.2: http:// favor.genohub.org/ GWAS summary statistics for neuroticism: https://ctg.cncr.nl/documents/p1651/ sumstats_neuro_sum_ctg_format.txt.gz Computational codes to generate Figures and a Supplementary Table: https://github.com/nbelon/ sumSTAAR-validation.

**Funding:** NMB GRS AVK IVZ YAT TIA received the funding from a budget project of the Institute of Cytology and Genetics (project number FWNR-2022-0020). NMB GRS AVK IVZ TIA received funding from the Russian Foundation for Basic Research (20-04-00464, https://www.rfbr.ru) YAT received the funding from the program "5-100 Best Universities" of the Ministry of Science and Higher Education of the Russian Federation (https://www. 5top100.ru/) The funders had no role in study design, data collection and analysis, decision to publish, or preparation of the manuscript.

**Competing interests:** The authors have declared that no competing interests exist.

individual phenotypes and genotypes as input data. To overcome these limitations, we propose sumSTAAR, a framework for gene-based association analysis using GWAS summary statistics. Our framework allows the user to arbitrarily define a set of the methods and functional annotations. Moreover, we adopted the methods for the analysis of genes with a large number of SNPs to decrease the running time. The framework includes the polygene pruning procedure to guard against the influence of the strong GWAS signals outside the gene. We also present new improved matrices of correlations between the genotypes of variants within genes, which now allows to include ultra-rare variants (MAF $< 10^{-4}$) in analysis.

This is a *PLOS Computational Biology* Methods paper.

## Introduction

Gene-based association analysis is an effective replacement of genome-wide association analysis (GWAS) in identification of rare genetic variants [1, 2]. Many gene-based methods have been proposed recently. Their power depends on the underlying genetic architecture that is rarely known in complex traits. Therefore, a combination of such methods could serve as a universal approach.

Among popular combined tests, SKAT-O was the first, for which the distribution of test statistics was analytically described [3]. For other combined tests, p-values were estimated empirically at the cost of dramatically increased computation time. The task to analytically combine the p-values obtained by different methods has been solved in the aggregated Cauchy omnibus test, ACAT [4]. This gave impetus to create a range of frameworks in order to one-by-one calculate a number of gene-based tests and then combine them by ACAT [5–9]. The frameworks differ from one another by the task, input data, methods used, and ways to include functional biological annotations. All these frameworks have a disadvantage: they are not flexible. They use the fixed set of methods, weights and combinations of functional annotations. Moreover, the majority of existing frameworks use individual phenotypes and genotypes as input data. Such data cannot be deposited in open-access databases, and so they are unavailable to a wide range of investigators. Recently, it was demonstrated that all popular methods of gene-based association analysis based on the linear regression models can use summary statistics instead of individual data [10]. Previously, we presented formulas for the wide range of association tests and implemented them in the sumFREGAT package [11].

The framework named STAAR (variant-set test for association using annotation information) stands out among others as a comprehensive and powerful tool that effectively incorporates SNP-weighting by allele frequencies, variant categories and multiple complementary annotations [6]. Here we propose the extended and modified version of the STAAR framework, which we called sumSTAAR. The modification concerns the input data: STAAR uses raw genotypes and phenotypes, and sumSTAAR uses GWAS summary statistics (effect sizes, standard errors, sample sizes etc.). Extension relates to the gene-based association analysis methods used: STAAR uses a fixed set of three methods, and sumSTAAR uses an arbitrary set including up to six methods.

The methods comprised by the sumSTAAR framework are modified in two ways compared with those previously described [11]. First, they involve a special algorithm for the analysis of large genes with >500 SNPs. Second, they use more efficient computational algorithms for

matrix operations. An additional empowering feature of the framework is the use of new LD matrices estimated on an extended sample: 265K instead of 503 individuals from the 1000G project. For the first time, due to these high-coverage estimates, it became possible to include the large amount of rare variants when analyzing summary statistics with a wide range of powerful gene-based methods. We also present the procedure of polygene pruning to guard against the influence of strong association signals outside the gene on the results of gene-based association analysis [12].

## Methods

### Gene-based association analysis

The sumSTAAR framework combines (a set of) the following methods: burden test (BT), SKAT, SKAT-O, aggregated Cauchy association test (ACAT-V), the tests using functional linear regression model (FLM) and principal component analysis (PCA). Variant-specific weights can be applied to all of these methods. We also introduced the probabilities of genetic variants being causal estimated using different functional annotations in BT, SKAT, SKAT-O, FLM, PCA and ACAT-V. All these modified tests and the parameters of their distributions are presented in S1 Text. The sumSTAAR() function (Fig 1) allows the user to arbitrarily choose a set of tests that differ in method, weighting function and probabilities of genetic variants being causal, calculate these tests, and then combine them using the aggregated Cauchy omnibus test, ACAT.

### Analysis of large genes

To decrease the running time for association analysis of genes with a large number of SNPs, we propose the following algorithm. Using thresholding technique, we divide all SNPs within a gene into two groups by p-values, which correspond to their weighted z-scores. Since multiple linear regression models include SNP-specific weights, we form SNP groups taking into account these weights. We used a threshold of 0.8, which was selected empirically (see below). We apply a chosen gene-based test to the group with the smaller weighted p-values and calculate the simple mean weighted p-value for another group. Then we combine p-values obtained for the two groups by ACAT. Obviously, this algorithm is an approximation to the chosen gene-based test, however it proves to be effective for the genes with more than 500 SNPs. We introduced it in SKAT, SKAT-O, PCA and FLM methods.

### Selecting the threshold

To choose the threshold, below which weighted p-values are considered as small when analyzing large genes, we performed an empirical assessment of the approximation on the material of summary statistics for neuroticism from UK Biobank dataset [13]. We calculated the approximated SKAT statistics using a range of values as threshold (from 0.05 to 0.95) on 2,103 genes having from 500 to 10,000 SNPs. For each threshold value, we measured the total elapsed time and calculated $R^2$ between the original and approximated SKAT statistics ($\log10(p$-values)). Since the approximated SKAT p-values deviated in both directions from original ones, we assessed both deviances using the formula

$$dev = \sqrt{\sum_i \left(\log_{10} P_{Ai} - \log_{10} P_{Oi}\right)^2}.$$

Here $P_{Ai}$ and $P_{Oi}$ are the approximation and original p-values for the $i$-th gene, respectively; $dev$ for conservativeness and inflation of the approximated test statistics was calculated using $i \in \{P_{Ai} > P_{Oi}\}$ and $i \in \{P_{Ai} < P_{Oi}\}$, respectively.

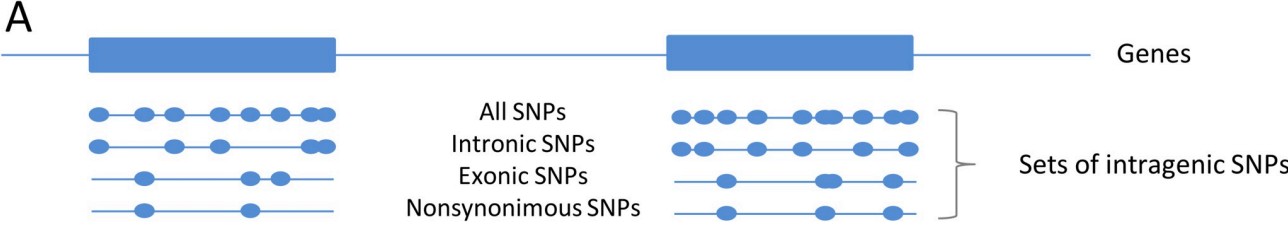

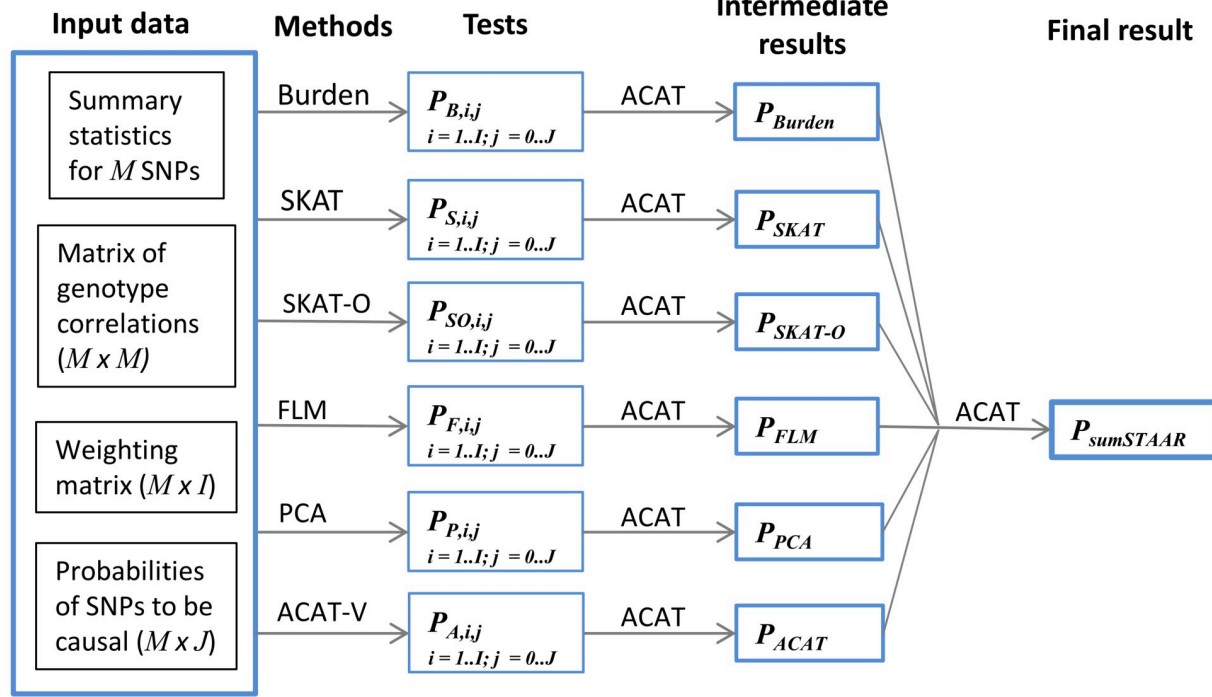

**Fig 1. Workflow schematic.** (A) Each set of SNPs (all, non-coding, exonic, nonsynonymous and others) is analyzed separately. (B) Input data for sumFREGAT include GWAS summary statistics (p-values and effect sizes), correlations between genotypes calculated using the same or reference sample, the matrix of weighting functions defined by the parameters of the beta distribution, the probabilities of SNPs being causal (e.g., estimated using different functional annotations http://favor. genohub.org/). The list of methods can comprise an arbitrary subset of BT, SKAT, SKAT-O, PCA, FLM, and ACAT-V. All methods use summary statistics as input. For each method, region-based association analysis is repeatedly performed using different combinations of the weighting functions ($i \in [1, I]$) and probabilities of SNPs being causal ($j \in [0, J]$). ACAT is used for combining the p-values obtained by each method under different weighting functions and probabilities, and then for combining the results obtained by various methods.

## Real data analysis

To test and evaluate the performance of sumSTAAR, we used two real data sets from the UK Biobank project [13, 14]. Sociodemographic, physical, lifestyle and health-related characteristics of the UK Biobank cohort have been reported by [15].

UK Biobank whole exome sequencing data and phenotype of the chronic ischaemic heart disease (IHD, ICD-10 code I25) contained 153,379 unrelated individuals (12,931 cases /

140,448 controls) with European ancestry (project #59345). We analyzed 110,538 variants covering 1,927 genes from chromosome 1 after the following filters: call rate $> 0.98$, MAC $\geq 5$ and MAF $< 0.01$. Sex, age and batch were used as covariates. These data were used for comparing the results of STAAR and sumSTAAR.

GWAS summary statistics for neuroticism contain information about the association for 10,847,151 imputed SNPs (MAF $> 0.001$ and INFO $> 0.9$) from a sample of 380,506 individuals and are freely available at https://ctg.cncr.nl/software/summary_statistics. Neuroticism levels were measured using the Eysenck Personality Questionnaire, Revised Short Form [16], consisting of 12 dichotomous items (0, 1). The quantitative trait was defined as a sum of 12 items (for details, see Nagel et al., 2018 [17]). These data were used for testing the new algorithm for analysis of large genes and for estimating the efficiency of different weighting functions and functional annotations defined via eight integrative scores (aPCs) from FAVOR v.2 (http://favor.genohub.org/) [6].

### LD matrices

The LD matrices of genotype correlations are required as input data for all packages using summary statistics. Using the UK Biobank resource under application #59345, we calculated Pearson correlation coefficients ($r$) between genotypes of variants within gene for 19,426 genes using 265,000 participants of European ancestry from the UK Biobank cohort [14] and LDstore software v1.1 [18]. Only variants with MAF$>10^{-5}$ and imputation quality $r^2>0.3$ were used for the calculations.

### Results

All three gene-based methods implemented in STAAR (BT, SKAT and ACAT-V) are available in the sumSTAAR framework. We analytically showed the equivalence of these tests between the frameworks (see S1 Text). We also numerically compared the results obtained in STAAR and sumSTAAR using simulated data (S2 Text) as well as UK Biobank exome sequencing data and phenotype of chronic ischaemic heart disease (S3 Text). STAAR implies an omnibus weighting scheme of combining multiple differently weighted tests (see S1 Fig). Using summary statistics, we reproduced this scheme in sumSTAAR and compared the results with those obtained by STAAR. As can be seen in S2 and S3 Figs, there is excellent agreement between the results obtained by two packages.

Then, we tested the new algorithm developed for analysis of large genes and picked up the threshold separating the two groups of SNPs in accordance with their weighted p-values. We tried to find a reasonable compromise between approximation accuracy and computation time. We observed the highest correlation between the original and approximated test statistics for threshold values ranging from 0.6 to 0.8 (Fig 2A). There was no prominent change in total elapsed time among these runs. However, the approximated test proved to be more conservative and inflation less frequent with increasing threshold values (Fig 2B). Therefore, to prevent an increase in false positive rate, we selected the threshold of the weighted p-value = 0.8 to separate the SNPs on two groups.

Using the neuroticism summary statistics, we estimated the accuracy and efficiency of the modified methods implemented in the new version compared to the version of sumFREGAT without modifications. Fig 3 shows a good agreement between the p-values obtained by two packages and a decrease in the running time when using the new modified version of the package. For SKAT, SKAT-O, PCA and FLM, the running time was decreased by 2.4, 10.5, 3.4 and 2.6 times, respectively. The most prominent effect was shown for the most popular SKAT-O method.

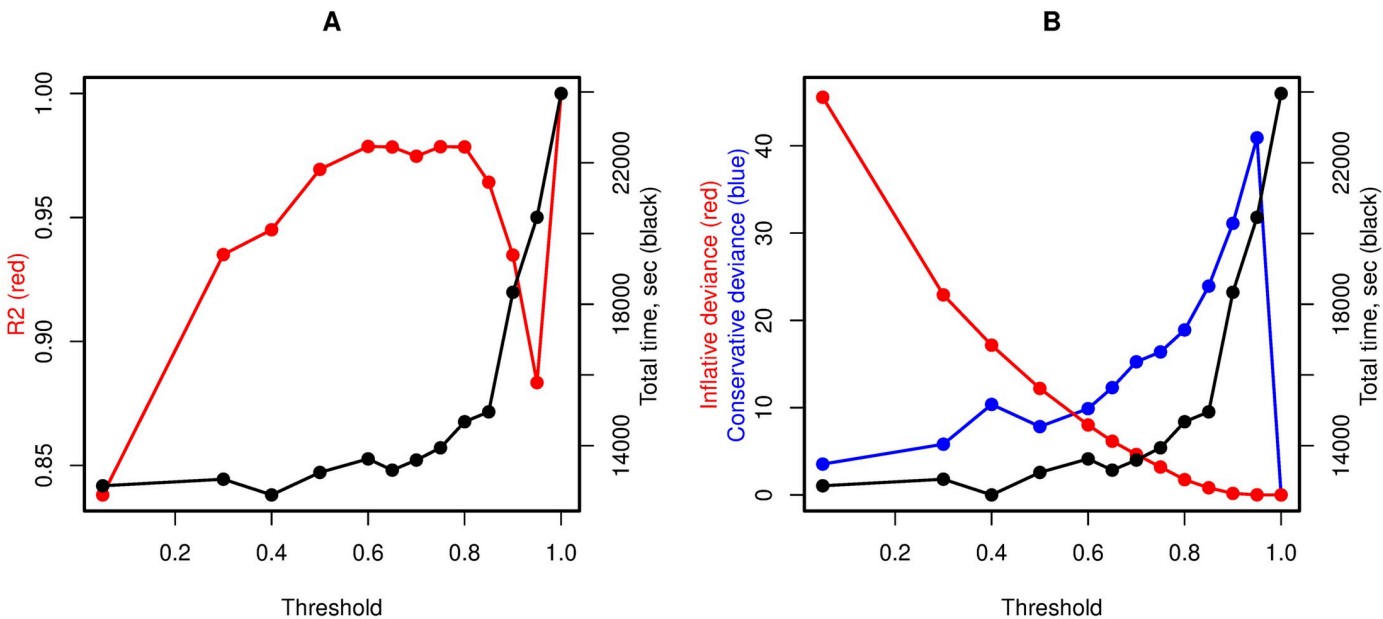

**Fig 2. Determination coefficient and deviances of approximated SKAT statistic related to the threshold value.** (A) Determination coefficient ($R^2$) between–log10(P value) of original and approximated tests shown in red. (B) Deviances indicating inflation and conservativeness of approximated test statistics compared with original shown in red and blue, respectively.

Within the framework, we introduce the new LD matrices for 19,426 genes estimated using genotypes of 265K participants of the UK Biobank project. The matrices contain information about 21,155,091 SNPs, with 17,142,006 (81%) of them having MAF < 0.01. For comparison, the widely used matrices of SNP-SNP correlations estimated on 503 European participants of the 1000G project include 4,544,901 SNPs, with only 707,862 (16%) of them having MAF < 0.01. The UK Biobank matrices, therefore, provide 4.65 times higher SNP coverage and 24 times higher coverage for rare variants. The matrices can be used in our or any other software together with summary statistics from samples of European ancestry. If available, other matrices calculated, for example, on Asian or African populations, can be used in our framework to analyze the corresponding samples.

For the polygene pruning procedure, we now publish an R-script to perform it step-by-step.

## Discussion

We developed a new framework for gene-based association analysis using summary statistics. This framework can include an arbitrary set of methods for gene-based association analysis, weighting functions and functional annotations used for the estimation of SNP probability being causal. Many of the methods used in the framework were adapted to the genes with large number of SNPs. This allows us to increase the computation speed of different methods by 2.4–10.5 times.

We compared STAAR and sumSTAAR and demonstrated the strong agreement of the results obtained by BT, SKAT, ACAT-V and their ACAT combinations. Our sumSTAAR framework, however, provides an opportunity to expand the range of methods with the fixed-effects models (PCA and FLM). High statistical power of these methods was previously shown

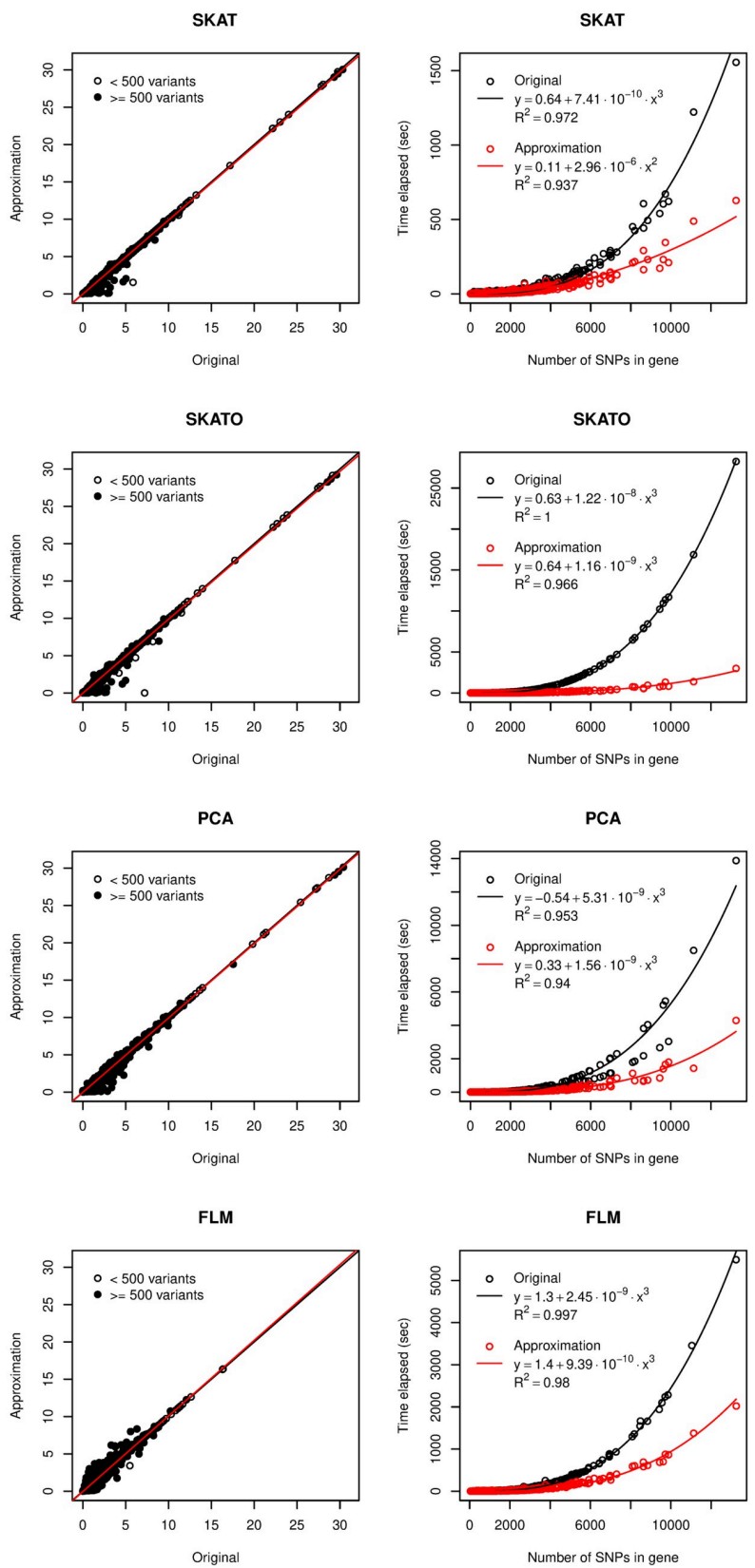

**Fig 3. Accuracy and running time of four gene-based methods for association analysis under approximation.** Each point represents a gene: 7,990 genes for FLM (genes that passed collinearity filter for 25 basis functions, see S1 Text for details) and 17,975 genes for other methods. Left panels show–log10(P values), red lines are regression lines and black lines represent one-to-one correspondence. On the right panels, lines represent the best-fitted polynomial functions.

for different simulation scenarios and real data [12, 19–22]. In addition, a method with random-effects model, SKAT-O, can be used instead of ACAT to combine BT and SKAT within the framework. Being more computationally intensive, SKAT-O nevertheless provides an optimal kernel-based combination of the methods when using the same weighting functions.

It is known that there is no universal or optimal method of the gene-based association analysis for any gene and trait (see for example, Wang et al., 2017 [23]). The more methods are used, the greater the chance of finding a causal gene. For example, in our previous study of neuroticism [12] the combination of the BT, SKAT, PCA and ACAT-V identified 190 genes, while the smaller set of BT, SKAT and ACAT-V would identify only 153 genes.

In addition to an extended set of methods, the power of the gene-based analysis can be increased by introducing various weighting functions and functional annotations. Using the neuroticism summary statistics for protein-coding variants, we showed that use of functional annotations and both weighting functions allowed us to identify a new neuroticism gene, *NARF* (see S4 Text). In total, using the additional PCA method increased the number of identified genes by 37, and additional weighting functions and functional annotations for protein-coding variants yielded one more gene. So, the effectiveness of our framework in this case can be estimated as 25% (38 / 153 * 100%).

Our framework allows researchers not only to increase the number of tests simultaneously included in analysis, but also to form their own alternative subsets of methods, weighting functions and functional annotations. Large number of tests can be time-consuming to compute, though it insures against selecting a method or weighing function that is not appropriate for a particular study.

To efficiently analyze the genes with large number of variants, we proposed a simple algorithm that implies subdividing the variants within a gene into two groups using a predefined p-value threshold. The running time decreases because the selected gene-based method applies only to the group of variants with lower p-values. For sequence kernel association tests, Lumley et al., 2018 [24] developed a fast approximation called fastSKAT. It does not directly limit the number of variants, but restricts the number of eigenvalues of genotype covariance matrix. Only 200 largest eigenvalues are included in the standard calculation of combined p-value; for the rest ones, the fast approximation is used. Due to the fixed size of the first group, fastSKAT running time for large genes grows with the square of $m$ (number of within gene variants) instead of the cube of $m$ for original SKAT. Unfortunately, the fastSKAT algorithm is specific to SKAT and cannot be applied to other methods of the sumSTAAR framework. Our algorithm is universal, though less efficient than fastSKAT because we fix the threshold for the p-values but not the size of the group. We analytically estimated the computation time to be half as less after applying our algorithm. This estimate assumes that running time increases as the cube of $m$ for all methods in the sumSTAAR framework and the p-values are uniformly distributed under the null hypothesis. The expected time reduction factor was therefore calculated as $1 / 0.8^3 \approx 2$, where 0.8 is the p-value threshold selected in our study. However, since we also updated some algorithms for matrix operations, the running time decreased by 2.5–3 times for all methods except SKAT-O that showed the most prominent effect of 10.5 times speed-up (Fig 3, right panels). Using the neuroticism as an example, we compared the p-values from the original and approximated methods for the large genes and showed a good agreement (Fig 3, left panels).

The sumSTAAR framework suggests using the polygene pruning procedure to guard against the influence of the strong GWAS signals outside the gene. It has been shown that a substantial share of gene-based association signals is inflated by these GWAS signals [6, 12]. To guard against this influence, the conditional GWAS summary statistics calculated using, for example, the GCTA-COJO package [25] can be used in sumFREGAT as input data. However, to calculate the conditional statistics, this type of analysis relies on the simple multiple regression with all the attendant limitations. For example, conditional SNPs should be in complete linkage equilibrium with each other and their number, therefore, cannot be large. The procedure called "polygene pruning" [12] represents an alternative way to reduce the effect of strong GWAS signals outside the gene. Polygene pruning results in exclusion of some variants within the gene being in LD with outside GWAS-identified variants from gene-based analysis. In essence, this procedure is analogous to the extreme weighting of within-gene SNPs based on their LD with outer GWAS signals. Polygene pruning way can be preferable when the set of within-gene variants is large or includes rare variants. Moreover, the classical conditional analysis is impossible to perform when genotypes of top GWAS signals are not available, while correlation structure sufficient for polygene pruning can be shared more easily.

Our framework can be applied to any summary statistics including those obtained by exome or whole-genome sequencing association analysis. We demonstrated such possibility comparing the results of STAAR and sumSTAAR obtained on the real exome sequencing data (see S3 Text). In practice, the application of our framework to these data is limited by the properties of existing reference LD matrices and summary statistics quality.

In principle, there are no restrictions to include variants with low MAC, even singletons with MAC = 1, in LD matrices, as we did in simulation experiment (S2 Fig). However, the ultra-rare variation is highly population-specific, and the robustness of their SNP-SNP correlations in the context of gene-based analysis was not yet estimated. Cross-population use of LD matrices for ultra-rare variants might, therefore, potentially lead to some uncontrolled errors. To bypass these problems, we ask the scientific community to publish genotype correlation matrices along with GWAS summary statistics. This would allow to perform the accurate population-specific gene-based analyses of the whole genome and exome sequencing data, as well as address the problem of correlation robustness for ultra-rare variants.

Another limitation of using the framework for sequencing data is the quality of summary statistics. Many GWAS tools are not designed to ensure unbiased Z-scores at low MAFs. If there is uncontrolled inflation in rare variants statistics, it will inflate the statistics of the gene-based analysis. For case-control association studies, we suggest using special tools that apply saddlepoint approximation correction for rare variants, for example SAIGE [26] or fastGWA-GLMM [27]. Moreover, if binary trait is analyzed, we suggest not to use variants with MAC $< 5$ since there is no robust algorithm to produce unbiased Z-scores, especially under unbalanced case-control design. For quantitative traits, more attention to departures from normality should be paid [28].

To conclude, we present sumSTAAR, a flexible and comprehensive framework that allows researchers to perform state-of-the-art gene-based analyses using GWAS summary statistics.

## Supporting information

**S1 Fig. Tests performed within the STAAR procedure.** Combined tests are shown in bold. (TIF)

**S2 Fig. Comparison of the results obtained by the STAAR and sumFREGAT packages.** Negative log10(p-value) were calculated in 10 simulations. The first three panels show the results for individual gene-based tests (Burden test, SKAT and ACAT) with two sets of

parameters for the Beta distribution and 11 variants of annotation weighting. The last panel presents–log10(p-values) for all combined tests. The regression lines are shown in red (overlap the lines of one-to-one correspondence).
(TIF)

**S3 Fig. Comparison of the results obtained by the STAAR and sumFREGAT packages.** The -log10 transformed p-value of each gene is shown. The first three panels show the results for individual gene-based tests (Burden test, SKAT and ACAT) with two sets of parameters for the Beta distribution. The last panel presents results combined across all tests. The regression lines are shown in red (overlap the black lines of one-to-one correspondence); 'r' is the correlation coefficient.
(TIF)

**S1 Text. Introducing the probabilities of genetic variants being causal and analytical equivalence of methods implemented in the sumFREGAT and STAAR package.**
(DOCX)

**S2 Text. Comparison of STAAR and sumSTAAR using simulated data.**
(DOCX)

**S3 Text. Comparison of STAAR and sumSTAAR using real exome sequencing data.**
(DOCX)

**S4 Text. SumSTAAR procedure in application to real data.**
(DOCX)

## Acknowledgments

The study was conducted using the UK Biobank resource under application #59345.

## Author Contributions

**Conceptualization:** Gulnara R. Svishcheva, Yakov A. Tsepilov, Tatiana I. Axenovich.

**Investigation:** Nadezhda M. Belonogova, Gulnara R. Svishcheva, Anatoly V. Kirichenko, Irina V. Zorkoltseva.

**Methodology:** Nadezhda M. Belonogova, Gulnara R. Svishcheva.

**Resources:** Anatoly V. Kirichenko, Yakov A. Tsepilov.

**Software:** Nadezhda M. Belonogova, Gulnara R. Svishcheva.

**Supervision:** Yakov A. Tsepilov, Tatiana I. Axenovich.

**Visualization:** Nadezhda M. Belonogova.

**Writing – original draft:** Tatiana I. Axenovich.

**Writing – review & editing:** Nadezhda M. Belonogova, Gulnara R. Svishcheva, Irina V. Zorkoltseva, Yakov A. Tsepilov, Tatiana I. Axenovich.

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
