## [Decision Letter · Decision Letter 0]

8 Jan 2022

Dear Dr. Belonogova,

Thank you very much for submitting your manuscript "sumSTAAR: a flexible framework for gene-based association studies using GWAS summary statistics" for consideration at PLOS Computational Biology.

As with all papers reviewed by the journal, your manuscript was reviewed by members of the editorial board and by several independent reviewers. In light of the reviews (below this email), we would like to invite the resubmission of a significantly-revised version that takes into account the reviewers' comments.

We cannot make any decision about publication until we have seen the revised manuscript and your response to the reviewers' comments. Your revised manuscript is also likely to be sent to reviewers for further evaluation.

Sincerely,

Andrey Rzhetsky

Associate Editor

PLOS Computational Biology

Ilya Ioshikhes

Deputy Editor

PLOS Computational Biology

Reviewer's Responses to Questions

**Comments to the Authors:**

Reviewer #1: Inspired by the STAAR framework, the authors extended their previous work in sumFREGAT to sumSTAAR. This paper is concisely written and easy to read. While there are some innovations in combining the two, e.g., new LD matrices, new algorithm for large genes, this reviewer does not find the contributions as significant. In addition, there seemed to be a lack of interest in answering any genetic problem using this new tool. There is no new findings other than the new tool agrees well with previously published ones and the speed is faster. The paper may strengthen if the authors can use this new tool to make significant genetic findings.

Reviewer #2: The manuscript describes a statistical framework in attempt to extend the STAAR method to analyze summary statistics data instead of individual genotypes data, and to expand its capacity to be able to incorporate additional models beyond what have been included in the STAAR framework. The extension to analyze summary statistics data is practical and will be useful when implemented properly. This reviewer appreciates the authors’ effort but has several major concerns in this study.

1. For gene-based and/or region-based rare variant analysis, the purpose is to analyze rare variants revealed in sequencing data, to increase power by aggregating multiple potentially causal rare variants that are not usually covered by GWAS data. The current study uses GWAS data for such statistical analysis, which is unlikely equally applicable to sequencing data. In order to make it rigorous so that it can be used by the community, it requires extensive simulation and application to real sequencing data, both of which are severely lacking in the current study.

2. For GWAS, many statistical methods are being developed or extended to summary statistics, which is a huge benefit to the research community given the easy accessibility of summery statistics. The adaptation of such strategy for rare variants uncovered by sequencing, however, is not straightforward. One reason is that the estimates of the effect size and standard deviation are not as accurate for rare variants, and another reason is that the LD among rare variants is challenging to estimate from an external reference panel given that rare variants are often subtly different among sub-populations. Both of the two aspects pose great challenges to the summary statistics-based rare variants analysis. The current study does not address the challenges, and it is not clear how these challenges may affect the analysis results. Extensive investigations are required to evaluate the impact of these issues.

3. The inclusion of additional models (e.g. functional linear models) requires additional evaluation to show the benefits of incorporating the additional models. Without such investigation, it is unclear whether the included models are able to increase power or not.

4. The separation of the SNPs in long genes is arbitrary. Although theoretically sound, in practice in real examples are needed to show that this is a sensible strategy.

Overall, as this tool is to be used by the researchers, it is critical that this tool is extensively evaluated to make sure that it delivers rigorous results under various scenarios. This is especially critical for rare variants, as it is well-known that there are unique challenges (e.g. point #1 and #2) and such rigorous evaluations are indispensable.

Reviewer #3: In this paper, the authors proposed sumSTAAR as a gene-based association analysis framework using summary statistics from genome-wide association studies (GWAS). The framework allows users to select candidate methods, weighting functions, and probabilities of genetic vairants being causal as introduced in the original STAAR framework. Overall, the proposed framework and data interpretations are justified.

I do have following questions and concerns regarding the manuscript.

Main comments:

1. My main concern is on the “Analysis of large genes” and “Selecting the threshold” in the Methods section. It is known that the original SKAT method (Wu et al. AJHG 2011) is scalable to perform SNP-set test for variant number of several hundreds to thousands. The authors are expected to demonstrate the advantage of computation for variant number more than 10,000. See FastSKAT (Lumley et al. Gen. Epi. 2018) paper as an example.

2. Although the proposed sumSTAAR framework uses GWAS summary statistics instead of individual-level data as input. Two main limitations should be discussed clearly by the authors in the revised manuscript:

(1) sumSTAAR could not be directly applied to large-scale sequencing studies, where most of the variants observed are extremely rare variants (e.g. 46% of the observed variants are singletons, see Taliun et al. Nature 2021). The LD matrices for 19,726 genes estimated using the genotypes of 265K participants of UK Biobank would not recover those variants.

(2) sumSTAAR could not be directly applied to multi-ethnic GWAS studies. As UK Biobank samples are mostly within European ancestry, the use of reference LD matrices calculated from UK Biobank participants may not be generalizable to other ancestries or multi-ethnic studies.

Minor comments:

3. In page 8 line 9, “aggregated Cauchy test, ACAT-O” should be “aggregated Cauchy association test, ACAT”. In addition, all the following appearance of “ACAT-O” should be “ACAT”, including page 8 line 11; page 9 line 21; page 10, line 2; Figure 1.

4. Following #3, in page 13 line 4, suggest changing “… and their ACAT-O-based combinations” to “… and their combinations” for clarity.

5. Following #3, in page 13 line 6-8, ACAT-O is a standalone variant-set test that combines the burden test, SKAT, and ACAT-V (Liu et al. AJHG 2019). It is shown that ACAT-O has better statistical power compared to SKAT-O. As such, it would be recommended for the authors to remove the sentence “In addition, …, compared with ACAT-O”.

6. In page 14 line 1-3, the conclusion sounds subjective to the broad readership of the journal. Suggest changing to “To our knowledge, sumSTAAR is a flexible and comprehensive framework that allow researchers to perform state-of-the-art gene based analyses using GWAS summary statistics”.

7. In page 11 line 23-25, since the authors utilized the STAAR package/tutorial and adapted the scripts based on https://github.com/xihaoli/STAAR/blob/master/docs/STAAR_vignette.html to facilitate the sumSTAAR vs STAAR comparison (https://github.com/nbelon/sumSTAAR-vs-STAAR-comparison/blob/main/sumSTAAR.vs.STAAR.R), it is recommended for the authors to acknowledge the STAAR package authors in the contributor list of the sumFREGAT package (https://cran.r-project.org/web/packages/sumFREGAT/index.html).

**Have the authors made all data and (if applicable) computational code underlying the findings in their manuscript fully available?**

Reviewer #1: None

Reviewer #2: **No: **The R code provided is only to generate supplementary FigS2. No computational code or data were provided for he main results including Figure 2 or Figure 3.

Reviewer #3: Yes

PLOS authors have the option to publish the peer review history of their article (what does this mean?). If published, this will include your full peer review and any attached files.

Reviewer #1: No

Reviewer #2: No

Reviewer #3: No
---

## [Decision Letter · Decision Letter 1]

29 Mar 2022

Dear Dr. Belonogova,

Thank you very much for submitting your manuscript "sumSTAAR: a flexible framework for gene-based association studies using GWAS summary statistics" for consideration at PLOS Computational Biology.

As with all papers reviewed by the journal, your manuscript was reviewed by members of the editorial board and by several independent reviewers. In light of the reviews (below this email), we would like to invite the resubmission of a significantly-revised version that takes into account the reviewers' comments.

We cannot make any decision about publication until we have seen the revised manuscript and your response to the reviewers' comments. Your revised manuscript is also likely to be sent to reviewers for further evaluation.

Sincerely,

Andrey Rzhetsky

Associate Editor

PLOS Computational Biology

Ilya Ioshikhes

Deputy Editor

PLOS Computational Biology

Reviewer's Responses to Questions

**Comments to the Authors:**

Reviewer #1: The authors have shown the immediate applicability of the sumSTAAR method using the neuroticism data. However, they were able to identify 1 additional associated gene, out of 190 previously found genes. It doesn't bring confidence to the significance of the method presented. And there is no guarantee that it always increase the number of genes identified. In particular, this reviewer agrees with Reviewer #2 that "it is critical that this tool is extensively evaluated to make sure that it delivers rigorous results under various scenarios." And the emphasis may be on evaluating this method extensively. It is hard to argue this method has been evaluated extensively on various scenarios based on the current revision.

Minor comment:

In Figure 1, it seems all 6 methods, BT, SKAT, SKAT-O, PCA, FLM, and ACAT-V, take in the exactly the same input. Is that the case? Since this paper emphasis the use of summary statistics, to avoid confusion it would be better for the authors to highlight which method uses summary statistics which doesn't.

Reviewer #2: The authors have addressed most of my initial concerns. Here are a few comments that could improve the quality further:

1) QQ-plot (of all p-values) needs to be provided to make sure the type I error is well controlled in each analysis. For example, QQ-plots for FigS4 and for each of the listed 8 combination scenarios as shown in Table S1 (using the p-values from all genes analyzed).

2) The URLs for the SNP-SNP correlation matrices of imputed genotype data of UKBB are not working and need to be fixed.

**Have the authors made all data and (if applicable) computational code underlying the findings in their manuscript fully available?**

Reviewer #1: Yes

Reviewer #2: **No: **

PLOS authors have the option to publish the peer review history of their article (what does this mean?). If published, this will include your full peer review and any attached files.

Reviewer #1: No

Reviewer #2: No
---

## [Decision Letter · Decision Letter 2]

5 May 2022

Dear Dr. Belonogova,

We are pleased to inform you that your manuscript 'sumSTAAR: a flexible framework for gene-based association studies using GWAS summary statistics' has been provisionally accepted for publication in PLOS Computational Biology.

Best regards,

Andrey Rzhetsky

Associate Editor

PLOS Computational Biology

Ilya Ioshikhes

Deputy Editor

PLOS Computational Biology

Reviewer's Responses to Questions

**Comments to the Authors:**

Reviewer #1: The authors have addressed all my concerns.

Reviewer #2: The authors have addressed all my concerns and I thank the authors for their revision efforts and recommend this manuscript for acceptance now.

**Have the authors made all data and (if applicable) computational code underlying the findings in their manuscript fully available?**

Reviewer #1: Yes

Reviewer #2: Yes

PLOS authors have the option to publish the peer review history of their article (what does this mean?). If published, this will include your full peer review and any attached files.

Reviewer #1: No

Reviewer #2: No

---

## [Editor Report · Acceptance letter]

27 May 2022

PCOMPBIOL-D-21-01971R2 

sumSTAAR: a flexible framework for gene-based association studies using GWAS summary statistics

Dear Dr Belonogova,

I am pleased to inform you that your manuscript has been formally accepted for publication in PLOS Computational Biology. Your manuscript is now with our production department and you will be notified of the publication date in due course.

With kind regards,

Zsofia Freund
